# Cellular and Molecular Immunity to Influenza Viruses and Vaccines

**DOI:** 10.3390/vaccines12040389

**Published:** 2024-04-07

**Authors:** Jane Kasten-Jolly, David A. Lawrence

**Affiliations:** 1Wadsworth Center, New York State Department of Health, Albany, NY 12208, USA; david.lawrence@health.ny.gov; 2Departments of Biomedical Science and Environmental Health Science, University at Albany School of Public Health, Rensselaer, NY 12144, USA

**Keywords:** influenza virus, vaccination, antibody, B cells, memory T cells, NK cells, macrophage, macroautophagy, epitope, idiotope

## Abstract

Immune responses to influenza (flu) antigens reflect memory of prior infections or vaccinations, which might influence immunity to new flu antigens. Memory of past antigens has been termed “original antigenic sin” or, more recently, “immune imprinting” and “seniority”. We have researched a comparison between the immune response to live flu infections and inactivated flu vaccinations. A brief history of antibody generation theories is presented, culminating in new findings about the immune-network theory and suggesting that a network of clones exists between anti-idiotypic antibodies and T cell receptors. Findings regarding the 2009 pandemic flu strain and immune responses to it are presented, including memory B cells and conserved regions within the hemagglutinin protein. The importance of CD4^+^ memory T cells and cytotoxic CD8^+^ T cells responding to both infections and vaccinations are discussed and compared. Innate immune cells, like natural killer (NK) cells and macrophages, are discussed regarding their roles in adaptive immune responses. Antigen presentation via macroautophagy processes is described. New vaccines in development are mentioned along with the results of some clinical trials. The manuscript concludes with how repeated vaccinations are impacting the immune system and a sketch of what might be behind the imprinting phenomenon, including future research directions.

## 1. Introduction

It has been observed that, in the case of vaccination targeting the influenza (flu) virus, the immune system can produce more antibodies (Abs) directed against virus strains first encountered by an individual rather than strains present in the vaccine [1,2,3]. This phenomenon was given the name “Original Antigenic Sin” (OAS) by Thomas Francis Jr. [4]. An early demonstration of OAS in humans was presented by de St. Groth and Webster [5]. Two monovalent flu vaccines (FM1 and SW) were administered to subjects divided into groups based on the presence of Ab as measured by hemagglutinin inhibition (HAI) titer to SW before vaccination. The results showed that the titer of the FM1 Ab after SW boosting was as high as that after vaccination with the FM1 vaccine, and the quantity of the Abs after SW administration was as high as that for FM1. A review about OAS by Monto et al. [6] described the persistence of Abs to a first infection with respect to a person’s age for three different pandemic strains of flu virus: ASw, PR8, and FM1. The data indicated that the older group, >30 yr, had high Ab titers measured by HAI to the 1931 swine strain, ASw; the 17–26-yr group had Ab titers to all strains studied; and the younger subjects, 4–10-yr, had Abs to only the most recent strain at the time, FM1. Children vaccinated against FM1 produced Abs only to FM1, but when they were vaccinated with ASw or PR8 they generated Abs to ASw and PR8 as well as high titers to FM1 [7]. However, adults vaccinated with FM1 produced comparable amounts of Ab to the older stains, suggesting decreased protection against the new strain of the virus. An illustration of the results of this experiment are shown in Figure 1.

**Figure 1 vaccines-12-00389-f001:**
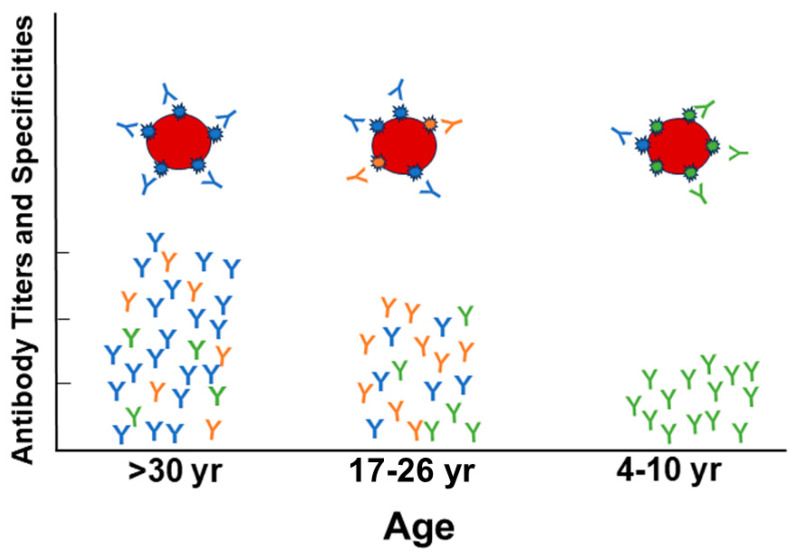
Depiction of the “original antigenic sin” phenomenon. The results of an experiment reported in 1957 by Davenport and Hennessy [7] showed direct evidence of the OAS phenomenon. Subjects were all vaccinated with a monovalent recent strain of the virus, FM1. Ab titers after vaccination measured by HAI indicated that the oldest age group, >30 yr, had more Ab generated against previous viral strains, ASw (blue epitopes) and PR8 (orange epitopes), than to FM1 (green epitopes). The young adult age group, 17–26 yr, had an even amount of Ab generated against each virus strain as measured by HAI. The youngest age group, 4–10 yr, had Ab generated only against the vaccine strain, FM1. Antibodies are matched to the colors of the epitopes.

Demonstration of protection against exposure to a pandemic flu strain was described by McCullers et al. [8]. The study found that individuals who were vaccinated with the 1976 flu vaccine had protection to the 2009 pandemic strain according to the HAI titers. Phylogenetic analysis showed that the 1976 vaccine strain (A/New Jersey/1976) bore similarities to the 2009 pandemic strain (A/California/7/09), and both had elements present in the 1918 pandemic strain. A more recent study described protection incurred by childhood exposure to the zoonotic strains H5N1 and H7N9, which indicated that exposure to the zoonotic viruses generated Abs that gave protection to novel hemagglutinin (HA) subtypes within the same phylogenetic group [9]. Therefore, if the virus strain antigenic differences incurred during childhood or recently by a new novel virus strain are large compared to current circulating strains, protection can be achieved, but if strain differences are small, strong protection could be compromised. This phenomenon was referred to as “immune imprinting” [9]. Immune imprinting was later explained as “the bias to use immune memory, independent of whether that immune memory was induced by the very first flu strain an individual is exposed to or an antigenically novel flu virus that an individual is exposed to later in life” [10].

This review presents a brief history of theories of antibody generation, followed by a discussion of how elements of the innate and adaptive immune systems interact to create high-affinity Abs that can protect against flu infection [11]. The subject of how vaccination can be affected by individual infection and immunization history will be covered. Also, human leukocyte antigen (HLA) diversity and cytokine gene expression variations will be suggested as contributors to diverse responses to flu vaccination [12,13]. The review will conclude with a review of efforts to create a more effective flu vaccine, followed by a discussion of how immune imprinting may still hamper these efforts. Discoveries about how repeated vaccinations influence immune responses will be presented as clues to what might be the mechanisms behind immune imprinting.

## 2. Theories of Ab Generation

Human immune responses to flu vaccinations were not readily explained by the early theories of Ab production proposed from 1930 to 1960. The first of these was the instruction model, which proposed that a foreign molecule, an antigen (Ag), would serve as a template for the Ab’s structure [14,15]. The second was the selective theory proposed by Burnet and Fenner [16], which proposed that it was not the Ag but the Ab that played a central role in determining specificity. The selection theory was later modified by Burnet [17], who suggested that the immune system generated a multitude of B cells with different hypervariable sequences upon encountering a specific Ag, and B cells with the best affinities respond by undergoing more clonal expansion. Most of the B cells produced during the expansion become plasma cells and produce Abs specific to the antigenic determinants (epitopes) of encountered Ags, but a few differentiate into memory B cells. Each foreign particle, like a virus, has multiple Ags and each Ag possesses epitopes that may stimulate specific responses in B cells with help from T cells to produce Abs with different idiotypes to each Ag’s epitopes. Adaptive immunity includes T cells with T cell receptors (TCRs), and B cell clones with high-affinity BCRs to an Ag’s epitopes may capture and process more Ag, providing longer cognate interaction with T cells (CD4^+^ Th cells) to obtain more cytokines, such as IL4, IL-5, and IL-6, for stimulation, proliferation, and differentiation to plasma cells. This might explain idiotypic differences in the levels of Abs to Ag epitopes. The selection theory was expanded by findings about immune responses to infections and/or vaccinations. In 1974, Niels Jerne presented his immune network theory [18]. This theory proposed that anti-idiotypic Abs could be generated against the variable domain (paratope and/or idiotopes) in an Ab’s hypervariable Ag-binding fragment (Fab), i.e., its Fab idiotype. The idiotype of a B cell’s epitope-specific receptors (its BCRs) are the same as the idiotype of the Abs that it produces, which recognizes an epitope of the Ag. In contrast, TCRs recognize a peptide of the Ag in union with the antigen-presenting cell’s (APC’s) major histocompatibility complex class I or class II (MHC-I or MHC-II) molecules for CD8^+^ and CD4^+^ T cells, respectively. Anti-idiotypic antibodies are made to Ab idiotypes, but Abs are not typically made to TCRs. These interactions would create a network of clones that have the potential to regulate immune responses. Infections and vaccinations generate a multitude of Abs in accordance with this immune network theory. An illustration of the immune network theory concept appears in Figure 2.

**Figure 2 vaccines-12-00389-f002:**
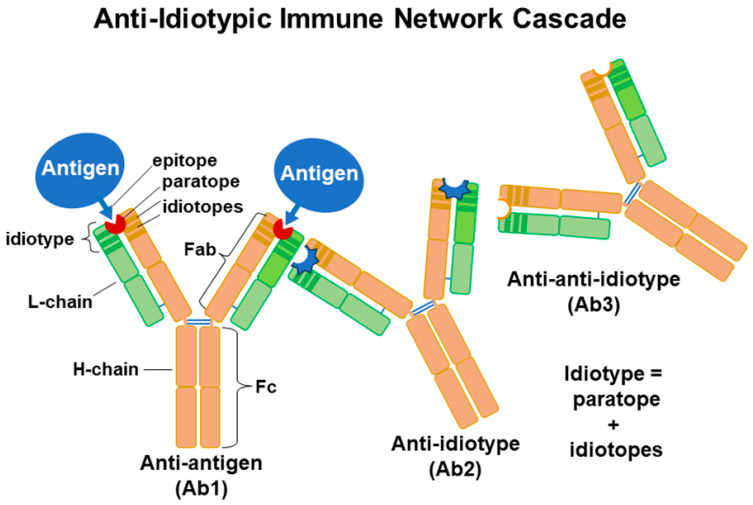
An illustration of the immune network theory. Anti-idiotypic Ab is generated against an idiotope or paratope present in an Ab’s Ag-binding fragment (Fab). Ab (Ab2) generated against the paratope might resemble the Ag’s epitope. An anti-anti-idiotype (Ab3) may be cross-reactive to the original Ag. In this manner, a network of idiotypes is established as a strengthening of the immune response towards a pathogen and possibly regulating Ab responses.

The first infection occurring during childhood would initiate the existence of memory B cell clones that survive from year to year based on the generation of anti-idiotypes, which may have sequences that resemble the viral epitope and anti-anti-idiotypes that might have specificities to the viral peptide sequences. Development of a method to identify idiotype-driven T-cell/B-cell interactions through sequences of BCR led to the identification of the corresponding TCR epitope [19]. The method defines patterns of TCR-recognized-epitope-motifs (TREMs) towards a viral Ag’s epitope(s) through analysis of BCR sequences. It is possible that this new method would be useful in the creation of more effective flu vaccines since past attempts at idiotype-based therapies have been disappointing [20].

## 3. Infection vs. Vaccination and Immune Response

Much has been discovered about the human immune response to flu vaccinations over the past seven decades, but some aspects of the OAS phenomenon remain a mystery, such as the association with early childhood infection as opposed to vaccination [21,22,23,24]. A childhood infection supplies a large concentration of Ag to naïve B cells for the development of memory B cells, as well as generating memory CD4^+^ and CD8^+^ T cells specific to the infecting strain’s Ags. Consequently, anamnestic responses, such as cross-reactive neutralizing Abs, antibody-dependent cellular cytotoxicity (ADCC), and cytotoxic T cells (Tc cells) will be diminished during subsequent infections [24]. New methodologies developed in the last few decades have provided results that shed some light on how the immune system reacts to a flu infection and the yearly flu vaccination [25,26,27,28,29,30]. The new information has provided some clues as to how OAS functions, and these recent findings have stimulated discussion towards the development of a universal flu vaccine [21,31,32,33].

### 3.1. Pre- and Post-Vaccination Abs to Flu Proteins

Despite a relatively high flu vaccination rate among humans, the flu virus still causes outbreaks every year due to mutations in amino acid sequence within the flu’s head group of the HA protein, which is the main target of neutralizing Abs [34]. These small changes in sequence have been described as a drift from the original viral sequence and can be shown diagrammatically as a phylogenetic tree [8], which illustrates how the flu’s HA may change a little each year and gradually move away from its parent HA protein. The genome sequencing of flu viruses taken from a collection of flu viruses in New York state and isolated over a period of several years reveals how flu has changed by small increments over time [35]. Due to these sequence changes on the head of the viral HA molecule, a new vaccine needs to be designed each year to immunize the human population against the strains that are anticipated to be in circulation. In addition to the HA sequence changes, the virus induces N-linked glycan changes that will also impact the generation of protective Abs against the HA protein [3], which demonstrated Ab responses against recombinant glycoproteins representing natural viral diversity as measured by ELISA. Human Ab responses against H1N1 and H3N2 infections revealed a broad range of responses and cogent evidence of OAS [3]. Flu vaccines have been termed trivalent because they contain three different virus types (2 A types and 1 B type): A/H1N1, A/H3N1, and B. More recently, quadrivalent vaccines have been generated containing two B flu types in addition to the 2 A types. A major finding from the study of Abs generated against the HA molecule from strains present in the yearly flu vaccine is that vaccines generate Ab diversification, which includes Abs binding to viral components not part of the vaccine [26,36]. A variety of methods have been employed to study Abs in pre- and post-vaccination serums. Included among these studies were B cell sorting and monoclonal Ab (mAb) development, ELISPOT, memory B cell activation, ELISA, HAI, and surface plasmon resonance (SPR) [3,25,37,38,39,40,41]. The HA present on the surface of the virus contains head and stem moieties. Most of the diverse Abs produced by vaccination are against epitopes present on the head of the HA molecule, since most mutations due to drift occur primarily in the head region of the HA molecule [25,26]. In contrast, the stem region sequences are more conserved. Abs generated against the stem region by certain individuals tend to be rather broadly reactive. These stem Abs may be more broadly reactive, but they will only bind weakly to the whole virus [42]. Unfortunately, revaccination against the same virus strain generates only Abs against the HA head moiety [43]. Abs generated during the yearly flu vaccinations have been found to bind to the strains present in the current vaccine, but also cross-react to strains absent from the vaccine (homosubtypic cross-reactivity). Table 1 gives the composition of flu vaccines from 2003 to 2017.

In our laboratory, this cross-reactivity has been demonstrated by SPR analyses performed using serum IgG from subjects who received the 2006/2007 vaccine and then the 2007/2008 vaccine the following year. Two study subjects who had never received flu vaccination and did not receive a vaccination until after their second blood donation, demonstrated low IgG binding to each vaccine at each pre-vaccine time-point. However, each subject had an increase in IgG binding to each vaccine at about four weeks post-vaccination (Figure 3A). Serum Ab at 14–15 days after vaccination from subjects who received the 2007/2008 vaccine displayed binding to all vaccines, including the 2008/2009 vaccine which had strains not present in the 2006 and 2007 vaccines (Figure 3B). All subjects who had received the 2006/2007 vaccine had Ab binding to this vaccine highest at two weeks post-vaccination; however, subjects with elevated Ab levels before vaccination generated less of a boost in the amount of Ab post-vaccination. The results indicate the high diversity of responses to the yearly flu vaccine within our human study population (Figure 3B). Control SPR for the 2006/2007 season was carried out by loading vaccine in equal proportions on flow-cells Fc2, Fc3, and Fc4. Pre- and post-serum IgG from 2007/2008 subjects was passed over all three of the vaccine-containing cells with the results showed very low variation in binding between flow cells (Appendix A).(Again, results from this chip indicated that elevated levels of flu-specific pre-serum Abs led to lower Ab responses after the vaccination. A study by Fonville et al. [44] followed individual responses to the yearly flu vaccine by correlating the H3N2 strain with HAI titers over a period of several years for 69 individuals; Ab landscapes were generated by correlating HAI titers to antigenically similar viruses and HA antigenic distance. This information could generate three-dimensional graphs whose appearance resembled a landscape after the data points were connected by smooth curves. A group of six individuals was followed from 2007–2012 and the resulting Ab landscape for each person showed that there was high variability in landscapes between individuals, but for each person the landscape shape was stable from year to year and had its own distinctive features [44]. The finding that each individual had a characteristic response to the vaccination year after year suggests that the genetic makeup of the individual might be affecting the immune response to the flu vaccine. Diverse human factors could include HLA variations, cytokine gene expression levels, and epigenetic environmental influences. In another study demonstrating high variability among individuals, subjects receiving the 2006/2007 vaccine had flu-specific IgG-secreting cells in the blood enumerated with ELISPOT and flow cytometry for 12 to 28 days following the vaccination [40]. It was found that peak frequencies for H1- and H3-specific Ab secreting cells (ASCs) were 26 ± 10% for H1 and 22 ± 17% for H3 of total ASCs. ASCs peaked between 5 and 10 days after the vaccination. Measurement of HAI titers over the 28-day period indicated that the HAI titers rose concurrently with the percentage of HA-specific ASCs. However, while the percentage of ASCs fell to near baseline levels around 28 days, the HAI titers remained elevated for most participants. One person did not have a measurable response to the 2006/2007 vaccine [40]. The high HAI titers found for the HA-specific IgG in this study peaked around day 12 and remained elevated over the 28 days of the study. This timing of 14–15 days post-vaccination generally corresponds to the peak of high flu-specific IgG found by our studies (Figure 3B). Further, like Haliiley et al. [40], Ab responses after vaccination showed a high amount of variation among study subjects (Figure 3B).

It has been shown that a yearly flu vaccination will generate Abs that are homosubtypic (against one type, like H1N1) or heterosubtypic (against all types, even those of animal origin). These cross-reactive Abs have been isolated and characterized by a variety of methods. A protein microarray of HA1 proteins from various flu strains was employed to identify memory B cells before and after vaccination [45]. The results revealed homosubtypic and heterosubtypic Ab production from memory B cells. Corti et al. [46] immortalized IgG-expressing B cells from four individuals after flu vaccination with H1 and H3 strains and found 20 heterosubtypic mAbs produced by the B cells. These mAbs were capable of neutralizing strains H1, H5, H6, and H9. Evidence of these cross-reactive neutralizing Abs generated by the yearly flu vaccine indicates that the vaccinations are providing a certain level of protection, depending on the individual, to most strains encountered during the flu season.

### 3.2. Study of the 2009 Pandemic Strain and Ab Response

The 2009 pandemic strain of H1N1 would be considered a shift in the flu’s yearly circulation [47]. A crystal structure study of the HA protein from the A/California/04/2009 H1N1 revealed that the HA had antigenic sites like flu circulating early in the 20th century [48]. Among the regions on the HA molecule favored for binding highly cross-reactive IgGs were the Sa and the Sb regions, and these were relatively conserved between the three strains studied (SC1918, PR8, and Brisbane07). For this reason, the older individuals were more protected from being infected than the young who did not have prior exposure to any of these strains. It is possible that activation of memory B cells specific to these conserved antigenic sites, Sa and Sb, could explain results from our study where IgG Abs isolated pre- and post-vaccination (12–15d) during the 2006 season displayed a relatively high boost in binding to the 2010/2011 vaccine, which included the 2009 pandemic strain, in three of the subjects, but two of the subjects did not display increased binding to the 2010/2011 vaccine (Figure 4).

The Ab responses to the 2010/2011 vaccine did not show a correlation with age, since the youngest (subject 141) had the highest boost in Ab and the two subjects with no Ab increase to the 2010/2011 vaccine (Subjects 094 and 064) were 51 and 40 years of age, respectively. The Brisbane07 strain, which contains the Sa and Sb sites, may have been already circulating during 2006; the three subjects that displayed a relatively high increase in binding to the 2010/2011 vaccine may have had exposure to the Brisbane07 strain and, therefore, had memory B cells directed toward the Sa and Sb conserved antigenic sites (Figure 4). How these memory B cells may lead to an increase in Ab against the pandemic strain has been suggested and illustrated by Guthmiller and Wilson [10]. Therefore, our results could fit the “antigenic imprinting model” where a new exposure against conserved epitopes results in antigenic imprinting of memory B cells where every new strain gets a place in the hierarchy. Analysis of Ab binding to a random peptide library from pandemic 2009 HA through display on the surface of yeast cells resulted in the identification of five antigenic regions, of which all but one was not present within the stem of the HA protein. Study of Ab binding to the pandemic 2009 H1N1 flu by yeast display and deep mutational scanning identified five single-domain Abs (nanobodies) that bound a highly conserved pocket in the HA2 domain of the pandemic HA [49]. These nanobodies bound with high affinity and inhibited virus membrane fusion and were considered neutralizing. Discovery of these highly conserved Ab binding sites within the HA stem provided hope that a universal flu vaccine could be developed. It was observed that the 2009 pandemic strain possessed a reassortment of elements from human, bird, and swine flu, and it was found that the HA epitopes for Sa and Sb could provide complete protection in mice vaccinated with a DNA vaccine containing coding sequences for these regions [50].

Broadly neutralizing Abs directed against the HA stem region have been detected in phage display libraries, and a study was performed to examine the extent of stem-reactive Abs generated by infection from the pandemic 2009 flu [47]. It was found that stem-reactive Abs increased significantly after infection with the 2009 pandemic H1N1, and HAI microneutralization values correlated with this result. However, the higher stem-reactive Ab levels after a pandemic virus infection returned to pre-pandemic levels at two years post-infection. The mAbs generated by plasmablasts obtained from individuals infected by the pandemic 2009 virus revealed that many of the neutralizing Abs were broadly cross-reactive to epitopes within the stem and head domains of multiple influenza strains [42]. It was speculated that these broadly cross-reactive Abs were produced by plasmablasts derived from activated memory B cells specific to conserved epitopes present on a variety of influenza strains. Monoclonal 70–1F02 was able to rescue all mice infected with pandemic 2009 H1N1, PR/8 H1N1, or FM/1 H1N1, while monoclonal 1009-2B06 was able to rescue mice infected with pandemic 2009 H1N1 or FM/1 H1N1. This suggested that these two mAbs had some therapeutic value and could be used to treat patients severely infected with a pandemic strain of flu.

The finding that the neutralizing cross-reactive Abs produced by individuals inflected with the pandemic 2009 virus that were derived from memory B-cells is reflected in findings about Ab generation after vaccination with the A/California/07/2009 (HINI) strain [37,51,52]. The mAbs derived from post-vaccination plasmablasts were screened for binding to the HA protein of the vaccine strain and other HAs, including H5 and H3. It was found that these mAbs could bind to more than one strain and showed a high degree of somatic hypermutation, indicating they were of memory B-cell origin. Several of the Ab showed a high degree of cross-reactivity and were found to bind to the HA stem region [37]. Seven highly cross-reactive neutralizing mAbs that bound to the HA stem region and, specifically, to the fusion peptide of HA2 have also been reported [51]. Broadly cross-reactive heterosubtypic Abs produced by vaccination with the A/California/07/2009 (HINI) strain protected mice from a lethal infection with the heterologous H5N1 strain [52]. These results have increased speculation that the broadly cross-reacting neutralizing Abs to the HA stem could be induced upon yearly vaccination with vaccines based on subtypes of HA not circulating among humans. However, repeated yearly vaccination with the A/California/07/2009 (HINI) strain as a component of the yearly flu vaccine from 2010 to 2016 has shown that Abs no longer bind the HA stem region, but only bind the HA head region [43].

### 3.3. CD4^+^ T Cell Memory to Influenza Virus Infection/Vaccination

Expansion of the T cells normally occurs following a viral infection. This increase in proliferation of pathogen-specific T cells is followed by the programmed cell death of the viral-specific T cells down to a few cells known as T memory cells [53]. Central memory T (T_CM_) cells can be identified in the peripheral blood by the presence of surface markers, including CCR7, CD45RO, HLA-DR, and CD38. CD4 memory T cell responses to flu infection have been reviewed [54,55,56]. Naïve and memory T cells become activated to become effector memory (T_EM_) cells by interaction with dendritic cells (DCs) (present in secondary lymphoid tissue) that present flu epitopes via class II HLA molecules on their cell surface. These T_EM_ cells begin proliferating and express Th1 cytokines, such as interferon (IFN)-γ. The T_EM_ cells transition to T_CM_ by upregulation of CCR7. The process of transitioning from T_EM_ to T_CM_ has been found to require IL-2 to sustain the memory cells during down-regulation of apoptotic signaling and upregulation of CCR7 [57]. These processes ensure that the rapid contraction of T_EM_ cells does not proceed to memory cell extinction but yields a population of stable CD4^+^ T_CM_ cells to protect against future pathogen assaults. Some of these memory cells become lung resident cells by upregulation of CD69 and CD11a. Other CD4^+^ T_CM_ cells enter the peripheral blood as CD45RO^+^ T cells and will be activated back to T_EM_ cells in response to vaccination [58]. A study performed in our laboratory during 2003 and 2005 indicated that memory (CD45RO^+^) CD4^+^ T cells in the peripheral blood decreased in number between 5 and 12 days after the vaccination (Figure 5A). Further study of the CD4^+^ T cells during 2003 indicated that the absolute number of CD4^+^/DR^+^/CD38^+^ T cells decreased in the blood between 5 and 12 days after vaccination with the 2003/2004 TIV vaccine (Figure 5B).

Presumably these T cells migrated to lymph nodes to be activated by APCs, such as DCs. A recent human study by Wilkinson et al. examined the potential utility of the pre-existing flu-specific CD4^+^ T cell memory cells [59]; in this study, humans were challenged with either H3H1 or H1N1 seasonal virus, and the CD4^+^ and CD8^+^ T cell responses were examined on day 7 and day 28. Results indicated pre-existing flu-specific CD4^+^ T cells, but not CD8^+^ T cells, responded to internal viral proteins and were associated with lower virus shedding and reduced illness. Therefore, pre-existing flu-specific CD4^+^ T cells will offer some protection from severe illness.

A study of memory T cell populations generated after stem cell transplantation in humans revealed that memory cells could not revert to naïve cells, but after several months post-transplant the thymus will start producing naïve T cells of donor origin [60]. An analysis of CD4^+^ naïve T cells, T_EM_, and T_CM_ in humans following the 2017/2018 vaccination in our laboratory indicated that at five days post-vaccination the number of naïve CD4^+^ T cells (CD45RA^+^/CCR7^+^) decreased, and the number of CD4^+^ T_EM_ (CD45RA^−^/CCR7^−^) increased slightly (Figure 5C). At two weeks post-vaccination, the number of CD4^+^ T_CM_ (CD45RA^−^/CCR7^+^) increased, and the number of naïve CD4^+^ T cells increased slightly over pre-vaccination counts, while the number of T_EM_ decreased slightly. At 63 days post-vaccination, all CD4^+^ T cell numbers were back to pre-vaccination levels. These results suggest that the thymus generates more naïve cells after the vaccination, since memory cells cannot revert back to naïve cells, as indicated by Rufer et al. [60]. Therefore, the increase in the naïve T cell counts post-vaccination at 14 days had to be due to naïve CD4^+^ T cells originating from the thymus. The results portray a scenario where naïve and T_CM_ cells are activated to become CD4^+^ T_EM_, which then proceed to become T_CM_ at 14 weeks post-vaccination. The number of flu-specific T_CM_ cells then declines with time, perhaps due to decreased antigen stimulation or some other mechanism. In the case of CD8^+^ memory T cells, a quick cytolytic memory response to a second infection would aid in control of the infection by decreasing the viral population. Most flu vaccines are composed of inactivated viruses and do not illicit a strong CD8^+^ T cell response. However, the vaccines promote generation of helper CD4^+^ T_CM,_ and they have been demonstrated to aid in the generation of pathogen-specific CD8^+^ memory T cells that can be activated by a secondary infection of the virus [56,61,62]. In this manner, the present vaccines could promote CD8^+^ T cell involvement.

Circulating CD4^+^ memory T cells will be able to help B cells develop into ASCs. With respect to influenza vaccination, HA-specific CD4^+^ memory T cells found in the blood are CXCR5^+^/PD1^+^/CXCR3^−^ [63]. These cells are now known to be of follicular origin and are termed CD4^+^ follicular helper (Tfh) T cells [64,65,66,67,68]. It has been established that Tfh cell production is promoted by a feed-forward loop by the transcription factors Bcl6 and Tox2 [67]. Figure 6 gives an illustration of the feed-forward loop proposed by Xu et al. [67].

Memory-specific CD4^+^ T cells against other membrane proteins, like M1 and NA, may also be effective in promoting a protective immune response [69,70,71]. However, memory cells towards the NP protein may be a negative factor in the Ab response [63]. Consequently, vaccines that promote CD4^+^ memory toward the virus membrane proteins, HA, NA, and M, could provide help to B cells that would lead to ASC secretion of cross-reactive Abs. The generation of neutralizing cross-reactive Abs to crucial flu membrane proteins could protect against future pandemic strains [72]. In a study by Wild et al. [68], pre-existing flu-specific CD4^+^ T cells were studied in the context of the yearly seasonal vaccine. Subjects were divided into three groups: 1. never vaccinated, 2. not vaccinated in the past 1 or more years, and 3. vaccinated in the previous year. By analyzing CD4^+^ T cell specificity to the HA_118-132_ epitope and HA IgG titers, it was found that CD4^+^ T cell and Ab responses to the vaccination were closely associated with a person’s infection and vaccine history. A strong “early” response was obtained for naïve participants and participants who had not been vaccinated the year before, while participants who had received the vaccine the year before had a somewhat delayed response. In the study by Wild et al. [68], transcription factors Tox and Tox2 were evaluated for their expression in circulating Tfh (cTfh) cells. It was found that Tox expression was higher in cTfh than in non-cTfh cells four days after vaccination. In the peripheral blood, cTfh cells can be distinguished from non-cTfh cells by their characteristic cell markers. Wild et al. [68] performed multiparametric cytometry for T cell markers, including those for Tfh cells. It was found that the flu vaccination promoted the proliferation of CD4^+^ T cells that displayed markers specific to Tfh cells and were ICOS^+^ and CD38^+^ [65]. The ICOS^+^/CD38^+^ subset of Tfh cells was associated with those individuals who were identified as “early” responders (day 4) after the vaccination. The Tfh cells are crucial for the development of humoral immunity because they provide help to B cells in germinal centers, via ICOS/ICOSL interaction, to promote antibody production [73]. Further analysis of Tfh early response and HA-specific antibody detection (day 28) were found to be directly associated. Both baseline levels of HA-specific IgG and the number of HA_118-132_-specific CD127^+^/CD4^+^ T cells also had an influence on IgG generation after vaccination. The fold change of Ab levels was lower in the presence of high HA-specific IgG and low HA_118-132_-specific CD127^+^/CD4^+^ T cells [68]. During the study of a universal flu vaccine construct created using a replication competent vaccinia strain (Wyeth/IL-15/flu), it was observed that memory CD4^+^ T cells were needed for early Ab production and these memory cells were primarily of the Tfh and Th1 type [66]. These findings seem to agree with the findings of Wild et al. [68] because, after applying dimension reduction with t-distributed stochastic neighbor embedding (tSNE) analysis to their data, it was found that four groups were generated where one and four were Th1-like and two and three were Tfh-like. Groups one and four corresponded with individuals who had a “late” response to the vaccination and groups two and three corresponded to individuals with an “early” response to the vaccination.

### 3.4. CD8^+^ T-Cell Response to Influenza Infection/Vaccination

Several studies have investigated the differences between the inactivated trivalent flu vaccine (TIV) and the live attenuated flu vaccine (LAIV) with respect to humoral and cellular responses [74,75,76]. The two vaccines are administered through different routes (TIV intramuscular and LAIV as a nasal spray) and, therefore, initiate CD8^+^ T cell responses through different antigen presentation pathways. LAIV presumably enters APCs in the mucosal tissue, undergoes replication, and promotes a strong innate immune response, whereas the inactivated virus is administered intramuscularly and promotes systemic immune responses. To measure cellular responses, peripheral blood mononuclear cells (PBMCs) were isolated pre- and post (28–42 day)-immunization from adults and children before and after receiving the 2004/2005 and 2005/2006 vaccines (LAIV or TIV) and were cultured for 17 h in the presence of the live fluA/Wyoming (H3N2) strain. After this incubation, the numbers of CD4^+^ and CD8^+^ T cells expressing IFN-γ were evaluated by flow cytometry [74]. It was found that the numbers of IFN-γ positive CD4^+^ and CD8^+^ T cells from the 2005/2006 season vaccination negatively correlated with the number of IFN-γ positive CD4^+^ and CD8^+^ T cells from the 2004/2005 season [74]. In the children, the flu-specific CD8^+^ T cell counts were slightly higher than those from adults. However, although neutralizing (HAI) Ab responses were negatively associated with baseline HAI titers for both vaccines, the LAIV vaccine had, overall, slightly lower protective Ab generated as measured by HAI. Study of the flu-specific CD8^+^ T cell population in the adults at 10 and 28 days post-vaccination with the 2004/2005 TIV formulation indicated no change in the number of CD8^+^ T cells expressing perforin, but it increased expression of CD27 at both 10 and 28 days post-vaccination [75]. Since CD27 is expressed by naïve and central memory cells, the significant rise in CD27 suggested that the number of naïve and central memory cells increased after vaccination. The absence of a change in perforin expression indicates a lack of CD8^+^ T cells with cytotoxic capability after the TIV immunization. No change was observed in the overall number of CD8^+^ T cells [75]. In our laboratory, cell counts for T cells, B cells, and NK cells did not significantly differ from pre-vaccination values at 5–12 days post-vaccination during the 2003/2004 and 2005/2006 seasons in adults vaccinated with the TIV seasonal vaccine (Appendix A). All TIV vaccines from 2003–2006 contained the A/New Caledonia (HIN1) strain, while H3N2 and B viruses differed between vaccines. Immunophenotyping of study subject volunteers during 2006 in our laboratory showed no change in absolute lymphocyte counts among the no-vaccination controls, but individuals who had received the LAIV vaccine in 2005 had a significant decrease in the number of CD8^+^ T cells in their blood (Figure 7A,B). Study subjects who had not received the LAIV vaccine in 2005 responded to the 2006/2007 vaccine with an increase in NK cell counts at 5–7 days after the vaccination (Figure 7C). Enumeration of lymphocytes at 3 and 14 days post-vaccination with the 2007/2008 vaccine indicated a decrease in CD4^+^ T cells and B cells at 3 and 14 days after the vaccination and a decrease in T cells at 14 days after the vaccination (Figure 7D). These results suggest that the composition of the vaccine will affect lymphocyte responses to the vaccination. Jegaskanda et al. [76] investigated the role of ADCC Abs and protection from flu infection in five cohorts as follows: (1) adults given monovalent H1N1pdm09 inactivated subunit vaccine (ISV), (2) adults vaccinated with monovalent H1N1pdm09 live attenuated vaccine (LAIV), (3) children vaccinated with seasonal ISV followed by seasonal LAIV, or with two doses of seasonal LAIV, (4) adults with community-acquired A H1N1pdm09 infection, and (5) adults experimentally challenged with influenza A (H3N2) virus. Results indicated that ADCC increased after natural infection and after vaccination with ISV in children and adults. Further, there was no increase in ADCC Abs after vaccination with the pandemic or seasonal LAIV vaccines. There was also no correlation between HAI titers and the amount of ADCC Abs [76]. Results from our laboratory for the enumeration of lymphocyte subsets at 5–7 days following vaccination with the inactivated TIV vaccine from 2010/2011 and 2016/2017 containing the pandemic strain, A/California/07/2009 (H1N1), revealed that the absolute number of the CD8^+^ T cells increased following the vaccination (Figure 8A,B). This suggested that the pandemic strain of the virus promoted a cytotoxic response immediately after vaccination. When the pandemic 2009 strain was removed from the yearly TIV vaccine in 2017/2018, the increase in CD8^+^ T cells was not observed after the vaccination, but T cell numbers did increase at 14 days post-vaccination (Figure 8C). Therefore, the type of vaccine and the strain composition play major roles in how the T cells respond to the flu vaccination [38].

### 3.5. NK Cells and Response to Influenza Virus Infection/Vaccination

It has been well established that NK cells are an important component of the innate immune response to flu infection, and several recent reports have shed further light on their role in humans [77,78,79,80,81,82,83]. During acute flu infection, the number of CD56^bright^CD25^+^ NK cells decrease in the peripheral blood, but after intramuscular flu vaccination these cells increased in the peripheral blood [82]. Moreover, acute infection led to decreased plasma levels of inflammatory cytokines, including IFN-γ, macrophage inflammatory protein (MIP)-1ß, interleukin (IL)-2, and IL-15. The data suggested that this NK cell subset was recruited into infected tissues to aid in clearing the virus. A comparison of the NK response to infection of human PBMC cultures with either a seasonal virus or a pandemic flu A strain revealed that the NK cells responded differently to the two viral strains [83]; the magnitude of IFN-γ expression by NK cells was higher for the Cal/09 pandemic strain. By mass cytometry, it was determined that the difference between strains was due to the amount of CD112 and CD54 on the surface of the targeted infected cells—in this case it was monocytes [83]. Cal/09-infected cells maintained a small amount of CD112 and CD54 markers on their cell surface, but the seasonal strain showed no evidence of CD112 and CD54 surface markers after infection. Therefore, live viral strains that lead to acute infection also promote up regulation of inflammatory cytokines and more cytotoxicity. However, it was noted that the inactivated viral strains in TIV do not decrease the presence of CD112 and CD54 on the surface of the monocyte cells [83]. A study of seasonal flu vaccinated mice showed that NK cells are responsible for decreasing illness after infection with a pandemic flu strain, such as A/California/4/2009 [84].

The importance of NK cells and their contribution to the development of a protective response to flu infection following vaccination with the human TIV yearly formulation has been the subject of investigation [85]. The study evaluated IFN-γ secretion by NK cells after vaccination with the 2003–2004 and 2005–2006 vaccine formulations [85]. Both vaccines contained the A/New Caledonia/20/99-like (H1N1) virus. The study included 10 subjects: 4 received either the 2003/2004 or 2005/2006 formulations, 1 received both vaccines, and 2 were unvaccinated controls. Each subject was drawn pre- and post-vaccination once/week for eight weeks. Lymphocytes were isolated by lymphocyte M separation and frozen after each sampling. Frozen lymphocytes were thawed and suspended in RPMI medium + 15% fetal bovine serum and the numbers of NK, CD4^+^, and CD8^+^ T cells were determined by flow cytometry. No changes were observed in the overall lymphocyte profile before and after vaccination for either vaccine. Lymphocytes were then placed in culture and stimulated with A/New Caledonia/20/99 whole virus and the cells expressing IFN-γ were quantified by flow cytometry. Additional cultures were stimulated with HA and M1 peptide mixes from the A/New Caledonia strain (79 of HA) and the A/Wisconsin/4754/94 strain (61 of M1). Results of the study indicated an increase in IFN-γ-expressing NK cells post-vaccination after stimulation with the whole virus strain, but stimulation with the HA or M1 peptides did not indicate any increase in IFN-γ expression.

The role of NK cells in the immune response to flu vaccination was investigated in several studies in mice and one in human cell lines or isolated human NK cells [86,87,88]. It was found that epitopes on the HA head could promote ADCC, which is carried out by NK cells for the clearance of virus-infected cells [88]. These HA epitopes were designated E1 and E2, and mice infected with the A/Hong Kong/415742Md/2009(H1N1)pdm09 virus developed Abs against these HA regions. However, it was found that although ADCC activity increased and the viral load in the lungs was slightly lowered, ADCC activity increased alveolar damage and increased mortality. It was concluded that the ADCC activity led to inflammatory cell infiltration into the lungs in the E1-vaccinated mice upon H1N1 flu challenge. Therefore, it was concluded that vaccines containing domains within the HA head that elicited an ADCC response would be detrimental rather than protective, and potential universal flu vaccines would have to strike a balance between the harmful and helpful effects of ADCC. Guillonneau et al. [87] studied cross-reactive immunity of cytotoxic CD8^+^ T lymphocytes toward conserved regions of the HA protein. It was found that including α-galactosylceramide (α-galcer) as an adjuvant component of the flu vaccine prompted NKT cells to increase expression of indolamine 2,3-dioxygenase (IDO). Although IDO acts as an immune suppressor, it promoted the survival of the cross-reactive memory CD8^+^ T cell population, thus increasing protection against challenge by a potential pandemic strain. Therefore, up-regulation of IDO-expressing NKT cells by inclusion of α-galcer as a vaccine adjuvant would be one method of increasing protection by the yearly TIV flu vaccine.

### 3.6. Viral Antigen Presentation by MHC Molecules—The Role of Autophagy

It has now been established that the cellular function of autophagy can deliver peptide-loaded MHC class II molecules to cell surfaces for presentation to CD4^+^ T cells [89]. This non-canonical function of macroautophagy machinery was first reported by Paludan et al. [90] for presentation of Epstein–Barr virus nuclear antigens to CD4^+^ T cells by EBV-positive lymphoma cells. For example, in the case of vaccination, extracellular antigens contained in the flu vaccine would enter the APC by LC3-associated phagocytosis (LAP) or, in the case of infection, newly synthesized viral proteins would be bound to ubiquitin and trafficked into lipid membrane vesicles coated with the autophagy-related gene protein 8 (ATG8) ortholog GABARAP. The bound-up ubiquinated viral protein cargo will be attached to p62, a scaffold protein in the phagophore, and then channeled through the autophagy pathway into the autophagosome and on to the lysosome for breakdown of the viral proteins into peptides before entering the MHC class II compartment (MIIC) where the peptides would be loaded onto the MHC class II molecules. This is a simplification of the process which includes many factors and autophagy-related proteins (ATG). ATG4 is a cysteine protease that functions to convert pro-factors into their active form. At present, there have been over 40 identified ATG. Some IgG Abs bound to virus may aid endocytosis by binding to the FcγRs of APCs. Some viral proteins from vaccinations also may bind to TLRs. Both FcγR or TLR binding can lead to recruitment of LC3 lipidation factors, including many ATGs. The lipidation of LC3 is stabilized by the NADPH oxidase 2 (NOX2). With the LC3-associated endosome or phagosome fusing with lysosome, the viral proteins are reduced to peptides and channeled into MIICs. Peptides from the degradation process would then be loaded on the MHC class II molecules which would be chaperoned to the cell surface. Antigen presentation through the autophagy machinery is illustrated in Figure 9.

Regulation of autophagy is associated with the availability of ATP, where depletion of ATP and increase in AMP, are signals of nutrient starvation and will increase autophagy. Vice versa, nutrient availability stimulates mTOR, the mechanistic target of rapamycin, which acts by phosphorylating the modulator of macroautophagy, ULK1, thereby decreasing the autophagy process.

It has become evident that autophagosomes selectively target their cytosolic cargoes via the assistance of autophagy receptors (ARs). Among these receptors is the TAX1-binding protein, also known as TRAF6-protein-1 (TAX1BP1/T6BP). Because the ARs contain ubiquitin and an LC3-binding domain, they can bind ubiquitinated proteins and traffic them into newly forming autophagosomes through interaction with LC3. Upon analysis of several ARs via gene silencing, it was found that T6BP was the only one capable of effectively enhancing presentation of autophagy-dependent antigens to CD4^+^ T cells [91]. It was found that T6BP functions in the presentation of both autophagy-dependent and independent endogenous processing and presentation of antigens by MHC class II molecules. The silencing of T6BP resulted in the rapid degradation of the invariant chain CD74 associated with the MHC class II molecule. Further study indicated that T6BP regulated degradation of CD74 through binding to the ER chaperone protein, calnexin (CANX). Therefore, T6BP is needed for regulation, through CD74 degradation, of processing and presentation of viral antigen by MHC class II molecules to CD4^+^ T cells.

It has been found that autophagy is necessary for maintaining the viability of both CD8^+^ and CD4^+^ T_CM_ [92,93]. In a model system where autophagy is deleted in mature cells, a comparison of autophagy function between CD8^+^ and CD4^+^ T cells reported that the elimination of autophagy greatly impacted the survival of the CD8^+^ T cells but did not affect the viability of the CD4^+^ T cells. Further study by injection of antigen experienced CD4^+^ memory T cells into naive mice indicated that autophagy-deficient cells were incapable of transferring humoral immunity. It was suggested that the autophagy-deficient CD4^+^ T cells suffered from the toxic effects of lipid overload and elevated mitochondrial activity, because of defects in mitochondrial function and elevated lipid in these cells [92]. A study of young vs. old subjects receiving vaccination for respiratory syncytial virus (RSV) demonstrated that memory CD8 T cells displayed decreased autophagy in the elderly (>65 yr) subjects [93]. It was found that this was due to a decrease in the autophagy-related metabolite, spermidine. Spermidine regulates autophagy via hypusination of eIFSA which then regulates the synthesis of TFEB, a transcription factor that contains two triproline motifs in humans. The addition of spermidine to PBMC cultures from older donors stimulated with anti-CD3/CD28 was found to increase memory CD8^+^ T cell viability through increased autophagy via a two-fold improvement in eIFSA and TFEB expression.

Presentation of epitopes from exogenous antigens and phagocytosed material on MHC class I molecules has been termed cross-presentation. This is an alternate path employed by DC and macrophages [94,95]. Most of the routes involving Ag capture for cross-prestation are through phagocytosis. Peptide loading onto MHC class I molecules can occur in the endoplasmic reticulum, or the phagosome followed by transport to the cell surface. In the case of DCs, only CD8^+^ DC are capable of cross-presentation in mouse cells. Here cross-presentation takes place in conjunction with the present autophagy pathway of the cells. In humans, CD1a^+^ but not CD14^+^ DC were capable of cross-presentation [95]. Like in the case of the mouse cross-presentation capable DC, the human CD1a^+^ DC displayed the presence of ubiquinated aggregates within the cell after activation by culturing overnight. Cell imaging results of the CD1a^+^ DC indicated that cross-presentation was being performed through the autophagy pathway.

Possible evidence of direct involvement of influenza A nucleoprotein in cross-presentation has been suggested through construction of a fusion protein containing the transduction domain of Tat from HIV type I with the C terminus, amino acids 301–498, restricted to HLA-B27 of influenza A NP [96]. It was demonstrated that the fusion protein was able to enter target cells and become processed correctly and the flu NP peptide was presented on Class I antigen. Although processing occurred within 1 h after the protein entered the cell, confocal microscopy results showed that most of the fusion protein accumulated in the trans-Golgi network. It was believed that it was the C-terminus peptide of influenza A NP that directed the fusion protein to the trans-Golgi network where it could be directed for proteolytic processing, perhaps via the autophagy process for peptide insertion into the Class I HLA molecule. The HLA B27 restricted peptide of the NP protein presented on B LDL bar cells was able to activate cytolytic CD8 T-cells, as demonstrated by ^54^Cr-release analysis.

### 3.7. The Role of Macrophages in Influenza Infection/Vaccination

Influenza A virus H1N1 is capable of infecting macrophages, but viral replication reaches a “dead end” in alveolar macrophages and new virions are not released. The interaction of flu and macrophages has been reviewed [97,98,99]. It has been determined that flu replication can continue in marrow-derived macrophages. Because the macrophage population is very diverse, replication has been studied in many types of macrophages and it has been found that viral replication is viral strain and macrophage subtype dependent [97]. The replicative ability of 16 HA subtypes was studied and it was revealed that H5N1 alone had by far the best ability to replicate in macrophages, due to its ability to overcome early blocks in the replication process. In humans, only the later blocks of replication have been studied well. These are the “dysfunction of nucleocytoplasmic trafficking and viral proteins” and “defective assembly and budding of infectious virions” [97]. Mature macrophages can be divided into two categories, M1 (classically activated) and M2 (alternatively activated), depending on the gene expression profile of cytokine expression after activation. Both M1 and M2 macrophages can be infected, but M2 are more susceptible to infection with H5N1 and CA/09. After infection, M2 macrophages become more M1-like and secrete the inflammatory cytokine TNF-α, which has been blamed for contributing to the hypercytokinemia and enhanced inflammation associated with H5N1 infection [97,98]. In the lung, the flu infects both epithelial and macrophage cells and reasons for flu replication blockage in macrophages compared to the absence of blockage in epithelial cells have been described [99]. Further, infected macrophage cells display increased phagocytosis of apoptotic epithelial cells, which aids in control of the infection.

In germinal centers (GCs) large macrophages had been identified over 100 years ago and were given the name tingible body macrophages (TBM) because of the presence of phagocytosed lymphocytes in their cytoplasm. TBM are huge cells and have been called “Gargantuan chameleons of the germinal center” [100]. Over the years it has been speculated that involvement of TBM in the immune response to infection was primarily limited to phagocytosis of B cells that had undergone apoptosis [101]. In mice it was noted that germinal center numbers of TBM decreased as the mice aged and TBM might be important for regulating the magnitude of the germinal center response to the infection [102]. This hypothesis was supported by an experiment with an (OVA)-specific T_H_ hybridoma that secreted IL-2 upon stimulation. These cells were cultured with OVA-bearing TBM, and IL-2 secretion did not occur but was inhibited. It was found that indomethacin added to the cultures could overcome this inhibition. Therefore, a role in downregulating the germinal center reaction was proposed for the TBM [103]. Impaired function of the TBM in their role of clearing apoptotic cells in the germinal center can result in autoimmunity. Experiments with Mer−/− mice demonstrated that Mer-deficient TBM had decreased ability to clear apoptotic cells and therefore, lead to an increase in antibody forming cells [104]. Mer is a tyrosine kinase receptor that functions by transducing signals from the extracellular matrix into the cytoplasm and, thus, can regulate cellular processes, such as phagocytosis. These data indicated that Mer on TBM has an active role in helping TBM clear apoptotic cells in the germinal centers. Recent studies of TBM have been performed by imaging their activities via movie recordings [105,106]. It was found that TBM are stationary cells that phagocytose the B cells with highly dynamic protrusions and accommodate final stages of the B cell apoptosis. Germinal center TBM are derived from bone marrow-derived precursors within lymphoid organs prior to challenge by infection or vaccination [105]. This macrophage differentiation process is driven by GC B cells. During immunization, TBM precursors in the follicle are activated by apoptotic B cell fragments and migrate to the GC where they appear as stationary TBM. In the GC the number of dead cells increases after the immunization and the TBM captures apoptotic fragments with long extended active processes followed by phagocytosis of the fragments [106].

### 3.8. Flu vaccine development

Presently, yearly flu vaccines are composed of inactivated virus of H1, H3, and two B strains. The vaccination goal is to generate neutralizing Abs against the head group of the HA protein based on the viral strains circulating during the current season [34]. Selected viral strains are grown in eggs and the process of producing the vaccine can take as long as six months. This runs the risk of changes in viral strains circulating by the time the vaccine is ready for distribution. In addition to these difficulties, the vaccine is, at best, 60% effective at protection, and any protection gained will wane by the end of the flu season. Therefore, past approaches to deal with hitting the moving target of circulating seasonal viral strains have not been adequately effective at warding off viral infection and viral spread and have left the world vulnerable to another possible influenza pandemic. For these reasons, in more recent years, there has been a surge in novel approaches aimed at new flu vaccination approaches. These approaches have been suggested in several manuscripts [33,107,108,109]. Many of these new potential vaccines have incorporated the findings mentioned previously to design their vaccines which involve the combination of viral epitopes and stimulants of certain host cell types, such as mucosal cells, to achieve better longer-lasting viral clearance capacity [110]. Development of a DNA-based vaccine that encoded conserved CD4^+^ T cell epitopes was used to vaccinate HLA-DR (HLA-DR4) transgenic mice [111]. The vaccine consisted of a DNA plasmid that carried the coding region for 20 virus epitopes. Viral proteins included were NP, M1, PA, PB1, PB2, and NS2. The immunized mice were challenged with a lethal dose of PR8, and 70% of the mice receiving the plasmid containing the viral epitope DNA survived, but only 10% of the mice receiving the empty control plasmid survived.

One approach to flu vaccination has been induction of mucosal immunity through an intranasal spray of attenuated live flu virus. By immunization in this manner, the effort is to activate the mucosal immune system to produce secretory IgA, which has a role in the prevention of respiratory illnesses [112]. However, nasal spray vaccines in use at present have not been able to increase protection over the injected seasonal vaccine. To increase the effectiveness of the intranasal spray vaccines, an assortment of substances, known as mucoadhesives, have been examined to increase residence time of the vaccine in the nasal cavity. These include cellulose, polyacrylate, starch, alginate, and gellan. Further, experiments with adjuvants that facilitate the transport of antigens through the mucus and epithelial cell layers are being performed. Among the transport enhancers are endotoxins (whose safety has been questioned), oil-in-water nano-emulsions, and mannitol among others. A vaccine manufactured by Berna Biotech incorporated an A-B toxin as an adjuvant in their nasal spray and found that it was linked to at least a 19 times higher rate of Bell’s Palsy after vaccination compared to the control group. Therefore, toxins are to be avoided as adjuvants. In addition, the viruses for vaccines are grown in eggs, and early efforts associated with vaccine production had vaccines containing several protein contaminants including endotoxins, which resulted in safety issues. A study of endotoxin content in various vaccines showed some direct association with endotoxin content and severity of systemic reactions [113]. Important for intranasal vaccines is the targeting of respiratory M-cells, cells of the mucosal associated lymphoid tissue (MALT), that can transport antigenic material to APCs. Since flu A-type viruses could adhere efficiently to the M-cells in vitro, it may not be necessary to add a facilitator for M-cell targeting to a flu vaccine preparation. Experimentation is ongoing to develop an efficient applicator for intranasal vaccine delivery. In the US, the AccuSpray device is currently being employed to administer the FluMist, but it can also be improved for delivery and dosage. In an attempt to develop a safer LAIV, a split segment vaccine was made by overlapping reading frames of M1 and M2 viral RNAs [114]. This split segment vaccine was safe when tested in mice and showed better efficacy than the current temperature-sensitive LAIV.

Although potentially dangerous, the addition of adjuvants to vaccine preparations has been found to increase the efficacy of flu vaccines. Increasing the cytolytic activity of CD8^+^ T cells proved to be helpful. Guilloneau et al. [87] found that adding α-galactosylceramide (α-galcer) to the TIV promoted NKT cell expression of IDO, which in turn promoted the survival of memory CD8^+^ T cells. Therefore, α-galcer would a potential candidate for an adjuvant in a seasonal flu vaccine. The adjuvant CAF01 was added to the seasonal split vaccine, and the adjuvanted vaccine was tested in Ferrets [115]. The results indicated that the adjuvanted vaccine was able to induce CD4^+^ T cell and antibody responses for protection against challenge by heterologous viruses. Strong immunogenicity was observed by adding a tocopherol-based oil-in-water emulsion adjuvant (AS03) to a prepandemic H5N1 virus vaccine [116]. A controlled phase trail in human adults demonstrated that the adjuvanted vaccine generated higher neutralizing antibodies against clade 2.2 (60.7% increase) and clade 1 (38.3% increase) than the nonadjuvanted vaccine. The adjuvant AS03 was found to be effective in a second vaccine trail for H5N1 [117]. It was determined that after the first immunization antibodies against the stem region of HA were cross-reactive and very protective. However, after booster vaccination, antibodies were generated primarily against the HA head region and were not as protective. It was speculated that the antibodies against the stem region from the first vaccination were blocking epitopes on the stem and inhibiting further antibody generation.

As indicated earlier in this article, it has been found that there are regions of the viral HA protein that are conserved, and that Abs directed against these regions can be neutralizing and afford protection. Therefore, many vaccine approaches involving the HA protein have been developed and have been reviewed [118]. These include chimeric HAs, mosaic HAs, computationally optimized antigens (COBRAs), headless HAs, and mosaic nanoparticles [118]. All these approaches showed promise when tested in mice and ferrets, but only the chimeric HA and the headless HA, i.e., mini-HA presented on ferritin particles, have undergone clinical trials [119]. A second ferritin particle vaccine (H1 ssF), which displays a stabilized H1 stem immunogen on a ferritin particle, has now undergone a phase 1 clinical trial [120,121]. The results indicated that the vaccine elicited a neutralizing antibody response against H1 viral subtypes. The conclusion was that the HA stem vaccine could induce broadly neutralizing B cell clones.

In addition to the HA protein, the influenza virus has RNA polymerase proteins, neuraminidase (NA), nucleoprotein (NP), matrix protein (M1), membrane protein (M2), nonstructural protein (NS1), and nuclear export protein (NEP). Vaccines have been developed that target several of these proteins as well [118]. The M2 protein provides a good candidate for a vaccine due to its conserved sequence between human and avian strains. Since M2 is an ion channel that is required for viral entry and egress from cells, Abs generated against M2 have the potential to be protective. Several methods have been employed for the development of vaccines containing M2. These include presentation on virus-like particles (VLPs), linking M2 to flagellin, and expressing it in a target cell with a DNA or mRNA vector. These vaccines have been successful in animals and have led to clinical trials which have indicated that M2 vaccines would be helpful in a combination approach with other vaccines, such as those for HA. The internal viral proteins M1 and NP are produced in relatively high amounts in infected cells and, therefore, peptides of these proteins are presented to T-cells via HLA molecules. Vaccines of M1 and NP combinations have been prepared by expression of the proteins by modified vaccinia Ankara (MVA) or by DNA or mRNA expression vectors. These vaccines have been in large Phase II clinical trials. Additionally, conserved peptides of the two proteins have been added to the standard yearly vaccine. The peptide addition approach has undergone a large phase III clinical trial.

Within the last four years, successful development of lipid-nanoparticle encapsulated nucleoside-modified mRNA vaccines has created a new pathway for vaccine production. Attempts to design a universal influenza vaccine are being based on this methodology [122]. One of these vaccines targets conserved sequences of the HA stem, M2, NA, and NP [69]. These vaccines were prepared for each protein singly and in combination. In mice, all animals survived a 50X LD_50_ challenge from an H1N1pdm virus strain when vaccinated with the combined protein vaccine, but mini-HA, M2, and NP alone showed only partial protection. Vaccination with the NA alone gave full protection even at the highest viral dose challenge by the H1N1 pandemic strain. However, the NA vaccine alone could not protect against heterotypic viral challenge, such as that caused by cH6/1N5 or H5N8. Another approach was to create a multivalent nucleoside-modified mRNA vaccine containing conserved HA sequences from all 20 known A and B influenza viruses [123]. It was found that Abs generated by the vaccine in mice were able to recognize both variable and conserved HA epitopes. Protection against an H1 pandemic strain challenge was not obtained when H1 HA sequences were removed from the vaccine. This result suggested that full protection could only be obtained if all 20 HA sequences were included in the vaccine.

Many flu vaccines are now undergoing clinical trials and these vaccines have been discussed in a recent review by Hu et al. [124]. Presented in this manuscript are descriptions of vaccines on six vaccine platforms. New possible adjuvants are mentioned along with new optimal routes of immunization. Several virus-like particles (VLPs) that function to display influenza antigens are discussed. In addition to two VLP-based flu vaccines that are in clinical trials, there are 22 in preclinical development. In addition to the ferritin nanoparticle, there are other nanoparticle platforms. Among these are one based on a pentameric lumazine synthase particle (FLuMos-v1) and another on an OVX313 nanoparticle. Viral vector-based vaccines are being developed to express flu proteins. Possible viral vectors include adenoviruses, poxviruses, herpesviruses, vesicular stomatitis virus, and lentiviruses. Four of the five flu vaccines currently undergoing clinical trials have been developed with the adenovirus platform. It is worth mentioning that there are currently five mRNA vaccines in phase III clinical trials and two of these are produced by Moderna and Pfizer/BioNTech.

### 3.9. Imprinting and vaccine boosting

A recent report has indicated that evidence of imprinting has been observed after repeated immunizations with the mRNA vaccine for SARS-CoV-2 [125]. It was found that booster immunizations with the bivalent vaccine, WA1/2020 and BA.5, produced a robust response against the original strain WA1/2020, but had a significantly lower response to the BA.5 variants. Subclass characterization of the Abs generated by the bivalent mRNA vaccine indicated that the major Ab was a IgG4 isotype. Therefore, during the course of repeated vaccinations for SARS-CoV-2, isotype switching had occurred from the initial IgG1 and IgG3 isotypes to IgG4. In terms of CD4^+^ T cells, this signifies skewing from a Th1 inflammatory response to a Th2 noninflammatory response. Another recent study concerning the flu vaccine studied DNA methylation patterns in a longitudinal analysis of yearly vaccinations [126]. In this study, it was found that repeated vaccinations lead to DNA methylation, primarily of genes associated with the signaling pathway for the pattern recognition receptor RIG-1, retinoic-acid-inducible gene-1. The genes found to be methylated and the influence on RIG-1 is shown in Figure 10.

Methylation of these genes resulted in increased function of RIG-1. The outcome of increased RIG-1 activity was an increase in transcription of type 1 IFNs (α/ß). Analysis of the cells associated with the repeated flu vaccinations has shown that mast cells are involved in some way [126]. A ligand for RIG-1 is 5’ ppp-dsRNA, and it has been found that incorporation of this ligand, as an adjuvant, with a flu vaccine markedly increased the generation of neutralizing high affinity antibodies in mice [127]. It was found that this ligand greatly enhanced activity within the germinal center by promoting Tfh cell induction. It is possible that repeated vaccinations are skewing responses toward Tfh2 function and transcription [65]. Activation of this process in humans will result in production of IgG2 and IgG4 as well as the involvement of mast cells. A representation of how repeated vaccination affects the immune response is presented in Figure 11.

### 3.10. Concluding remarks

First exposure to the flu virus usually occurs during childhood and, in most cases, has a high impact on the body. During the infection, the body attempts to bring the infection under control by an interaction between the innate and adaptive immune systems. Depending on the individual’s cytokine expression levels the degree of inflammation generated by the infection could be rather high. One of the ways the body would attempt to bring down the inflammation and reduce the virus load might be to initiate epigenetic modifications to the DNA. These modifications might be designed to trigger a stronger response by the innate immune system to infection by a similar virus. The outcome is that, during vaccination with drifted virus strains, a higher antibody response is observed in the first strain encountered, perhaps due to the epigenetic modifications. Further, the first viral infection will generate long-lasting T and B cell clones that will be directed against conserved viral protein sequences. These clones will be activated during infection by a shifted virus, like the pandemic 2009 virus. Therefore, the imprinting phenomenon would be the result of the interplay between the innate and adaptive immune system and an individual’s genetic makeup.

## 4. Materials and Methods

### 4.1. Reference Search Method

The databases of OVID and PubMed were searched using general terms, such as “influenza virus infection” or “antibody response to virus infections”. Papers found were then selected based on the relevance to the manuscript subject matter and more subject-specific terms were searched, such as “CD4 Tfh memory cells and B cells”. The search engine “Bing” was also helpful in that it provided a list of more recent papers when we downloaded a manuscript found using an OVID or PubMed search. All references were selected based on appropriateness to the subject matter of the present manuscript.

### 4.2. Characteristics of the Study Populations

Peripheral blood was obtained from 11 consenting human donors, 7 females and 4 males, in 2003 and in 2005, 18 consenting human donors, 12 females and 6 males before and 7–12 days after the flu shot. In 2003, donors were immunized by intramuscular injection with a vaccine of the following composition: A/New Caledonia (H1N1)/Panama (H3N2), B/Hong Kong, and in 2005 the vaccine composition was A/New Caledonia (H1N1)/New York (H3N2) B/Jiangsu. The 2006/2007 season study had 24 females and 12 males enrolled. The average age was 47.5 yrs. The age range was 25 to 67 yrs. The 2007/2008 season study had 15 females and 10 males enrolled. The average age was 49. The age range was 26 to 68 yrs. The 2010/2011 study had 3 individuals enrolled, 2 males and 1 female. The 2016/2017 study had only 1 male and 1 female, and the 2017/2018 study had 3 males and 2 females. The median age for these smaller studies was 45.1 and the age range was 22–72. All study participants were either students or New York State Department of Health employees. Study participants completed the questionnaire and signed the consent form as part of the NYS IRB-approved protocol 98–108. Blood samples were drawn by licensed phlebotomists. Immunizations were intramuscular, and the vaccine compositions are listed in Table 1.

### 4.3. Isolation of Human IgG

Blood drawn into red-topped vacutainer tubes from BD Biosciences was allowed to clot at room temperature followed by centrifugation (300× *g* for 10 min at 4 °C) to pellet the clot and obtain the serum. The serum was aliquoted into freezer vials (500 µL each) and stored at −70 °C. Pre- and post-12–14-day serum IgG was isolated via the NAB G spin column system by Pierce Biotechnology according to the manufacturer’s instructions. The concentration of IgG Ab was determined with a nanodrop spectrophotometer ND-1000 (Thermo Scientific, Waltham, MA, USA).

### 4.4. Antibody Binding Analysis by Surface Plasmon Resonance

A BIA3000 instrument (Pharmacia Biotech, Uppsala, Sweden) was employed for the SPR measurements. Vaccines were attached by amine coupling at similar RU (4500–5000) to flow cells 2 (2006/2007 formulation), 3(2007/2008 formulation), and 4 (2008/2009 formulation) of a Biacore CM5 chip. Flow cell one was a blank surface control. IgG (500 µg/mL) was injected (40 µL) over the CM5 chip at 10 µL/min. Running buffer was degassed and filtered HBS-EP, pH 7.5 plus soluble 1 mg/mL CM dextran, and the chip surface was regenerated with two pulses of 10 µL each of 10 mM glycine, pH 1.5, at a flow rate of 20 µL/min. Response unit (RU) readings were taken at 15 sec after completion of the sample injection and about 100 s later before regeneration of the chip surface. All RU readings were programmed to be measured mechanically by the instrument software. No manual measurements were made to avoid error due to time discrepancies. A second CM5 chip contained 2006/2007 vaccine on flow cells 2, 3, and 4 in nearly equal amounts (4000–5000 RU). This chip was made as a control to be sure that each flow-cell was responding to the sample in a roughly equal manner.

### 4.5. TBNK Immunophenotyping

Blood samples were collected between 0915 and 1130 h in heparin or EDTA vacutainer tubes and were stored at room temperature no longer than 24 h before processing. Immunophenotyping for T, B, and NK cells was performed using the 4- or 6-color MultiTest reagents from BD Biosciences (San Jose, CA, USA). The whole blood was stained in TruCount tubes according to the manufacturer’s protocol. Stained samples were run on BD FACSCalibur or FACSCanto instruments using MultiSet software (v1.1.1) or FACSCanto (v2.0) clinical software, respectively.

### 4.6. Memory Cell Analysis

In 2003, T-cells were analyzed for the presence of CD45RA and CD62L using the four-color combination CD45RA/CD62L/3/8, according to the fluor order FITC/PE/Per-CP/APC from BD Biosciences (San Jose, CA, USA), and for the presence of CD38 and HLA-DR using the four-color combination 4/CD38/3/HLA-DR. In 2005, 4 color combinations were prepared for CD45RA/CD62L/3/8 and HLA-DR/CD45RO/3/4 from single color reagents purchased from BD Biosciences (San Jose, CA, USA). Cells were stained as for the immunophenotyping procedure above and were run and analyzed under identical settings, between 2003 and 2005, using the CellQuest software (v3.3) on the FACS Calibur flow cytometer. Absolute cell counts obtained from the MultiSet software were employed to calculate absolute cell counts of subsets bearing the CD45RA, CD62L, CD38, or HLA-DR surface markers. In 2017 for naïve and memory T cell determinations, the Human Naïve/Memory T Cell Panel from BD Biosciences (San Jose, CA, USA) was employed and EDTA was used as an anticoagulant. Each tube for cell labeling contained 100 µL of blood and 5 µL each of labeled antibodies, Alexa-fluor 647-anti-human CD197(CCR7), APC-H7-anti-human CD3, PerCP-Cy5.5-anti-human CD4, and FITC anti-human-CD45RA. Tubes were vortexed briefly to mix and were incubated at room temperature for 30 min in the dark. After labeling, the red cells were lysed with 1X Pharm Lyse and the leukocytes were washed 3X with phosphate-buffered saline. Labeled cells were then resuspended in 1 mL of 1 X FACS lysing solution and run on the FACS Canto flow cytometer using the FACS Diva software (V6.1.3). Gating strategies for analyzing naïve and memory flow data are shown in the Appendix A.

## Figures and Tables

**Figure 3 vaccines-12-00389-f003:**
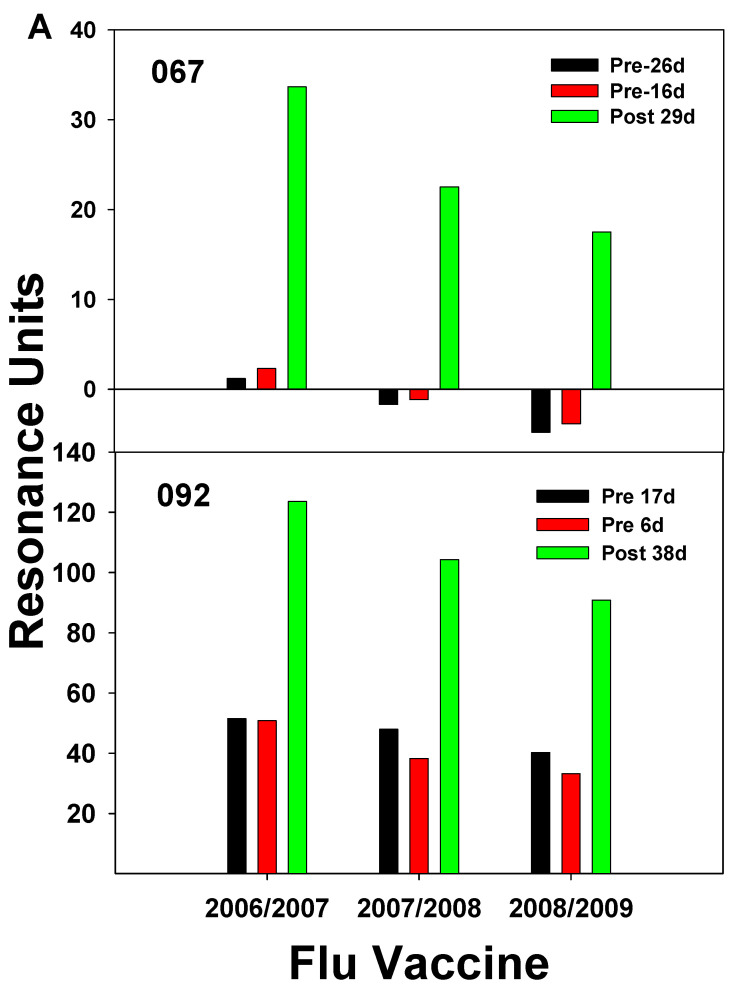
Homosubtypic cross-reactivities of IgG serum Abs to different flu strains are revealed by SPR measurements. Pre- and post-13–15-day serum IgG Abs were isolated with a NAB spin column according to the manufacturer’s protocol. IgG binding to three different vaccine preparations was evaluated with a BIA3000 instrument equipped with a CM5 chip coated with three different vaccines as described in the Section 4. (**A**) Pre-2006 vaccination IgGs at two time points followed by vaccination with the 2006/2007 vaccine (**B**) Pre- and post-13–15 days for the 2007 vaccination.

**Figure 4 vaccines-12-00389-f004:**
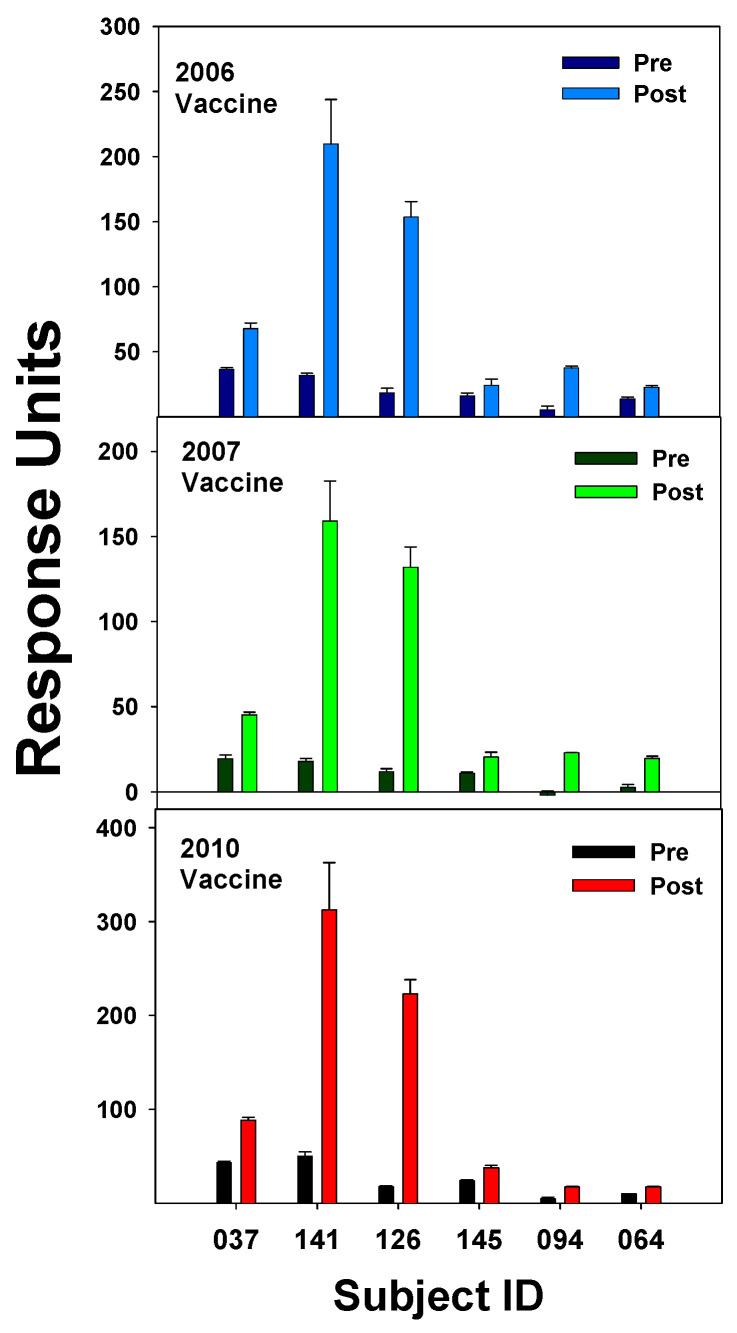
Capability of several subjects from the 2006 study group to generate Ab against the 2010–2011 vaccine containing the pandemic strain. The Ab (IgG) was isolated from serum before and after (12–14 days) the 2006 vaccination. IgG binding was evaluated by surface plasmon resonance as described in the Section 4.

**Figure 5 vaccines-12-00389-f005:**
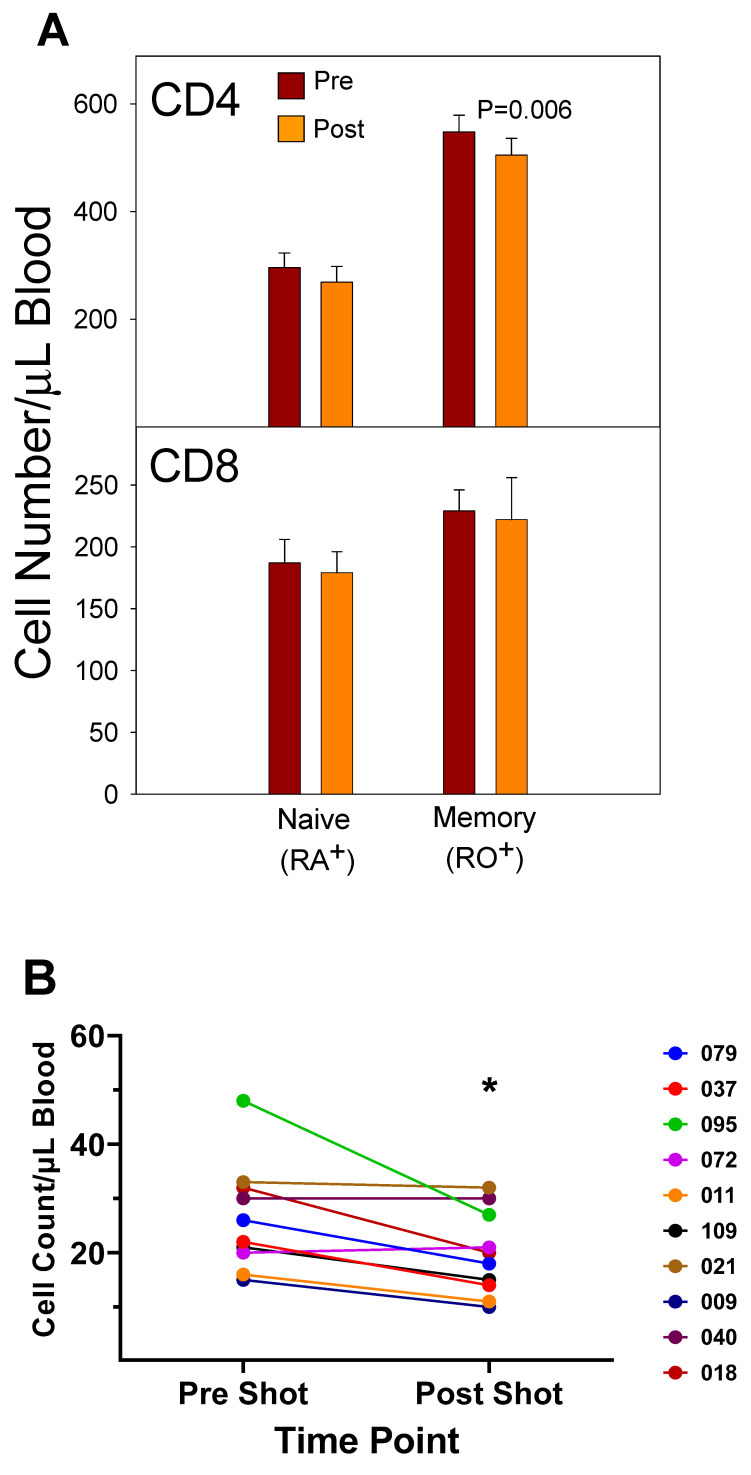
The number of CD4^+^/CD45RO^+^ (memory) T cells decreased in the blood at 5–12 days after the influenza vaccination. Memory and naïve (CD45RA^+^) T cells were analyzed by flow cytometry during the 2003 and 2005 studies as described in the Section 4. (**A**) Comparison of pre- and post-naïve and memory CD8^+^ and CD4^+^ T cells (combined 2003 and 2005 study) (N = 29). Analysis performed by CellQuest software v3.3, (Appendix A). (**B**) Pre- and post-CD4^+^/CD38^+^/DR^+^ T cells from 2003 study (N = 10). Analysis performed by Cell Quest software v3.3, (Appendix A4). (**C**) CD4^+^ naïve and memory T cell percent pre-vaccination counts for the 2017 study (N = 5). Central Memory (CM), Naïve (N), Effector Memory (EM). Analysis performed by FlowJo software v1.7.1_CL, (Appendix A). Significance was determined by a paired Student’s *t*-Test, “*” significant at *p* < 0.05.

**Figure 6 vaccines-12-00389-f006:**
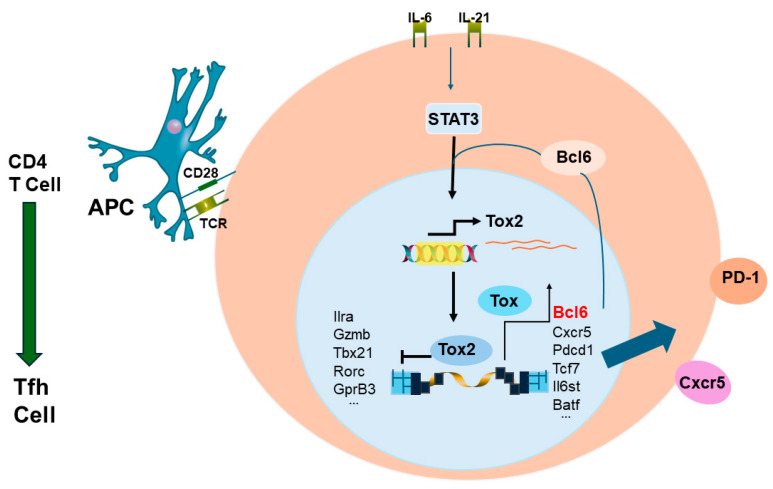
A depiction of the role of Tox2 in generation of CD4^+^ Tfh cells. Bcl6 is the essential lineage transcription factor in Tfh cells. The role of the transcription factor Tox2 is to drive Bcl6 expression and Tfh development. Tox2 binds to loci associated with Tfh cell differentiation and function. Binding of Tox2 to the chromatin increases chromatin accessibility at loci associated with Tfh differentiation. The Tox2-Bcl6 axis creates a transcriptional feed-forward loop that promotes Tfh development as discovered by Xu et al. [67]. In this scheme, IL-6 and IL-21 bind to antigen stimulated CD4^+^ T cells and induce STAT 3 activation. STAT 3 then promotes expression of Tox2, which then promotes transcription and expression of Tox. It is the Tox that generates the feed-forward loop effect by its interaction with Bcl6. Interaction of Tox2 with the chromatin will promote the expression of factors associated with Tfh cells and block the expression of factors associated with other types of CD4^+^ T cells.

**Figure 7 vaccines-12-00389-f007:**
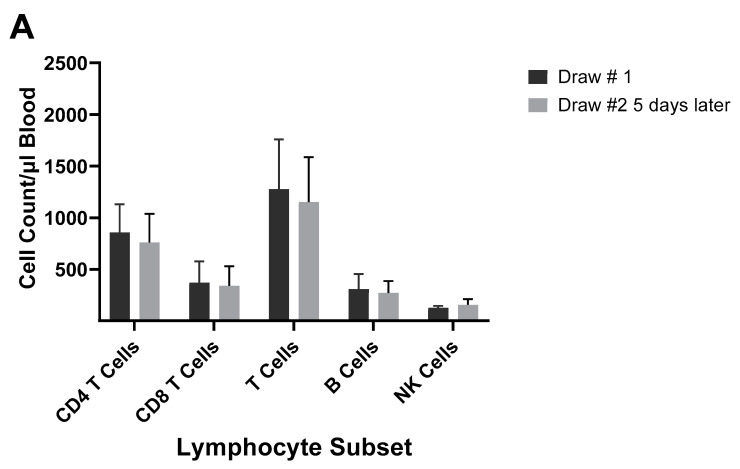
Enumeration of the lymphocyte subset counts before and after the 2006 and 2007 vaccinations revealed that vaccination history and vaccine composition affect lymphocyte subset responses following the vaccination. Lymphocyte counts were obtained by the 4-color MultiTest method by BD Biosciences with BD clinical software v1.1.1 on a BD FACS Calibur flow cytometer. All were performed according to the manufacturer’s protocol. Immunophenotyping results passed the criteria designated by BD Biosciences software v1.1.1 for accuracy, i.e., lymph sum and count ranges between duplicate samples. (**A**) Nonvaccinated controls (N = 5). (**B**) Subjects who received FluMist in 2005 (N = 3). (**C**) Subjects vaccinated with TIV 2006/2007 (N = 20). (**D**) Subjects vaccinated with TIV 2007/2008 (N = 22). Significance was determined by a paired Student’s *t*-Test, “*” significant at *p* < 0.05.

**Figure 8 vaccines-12-00389-f008:**
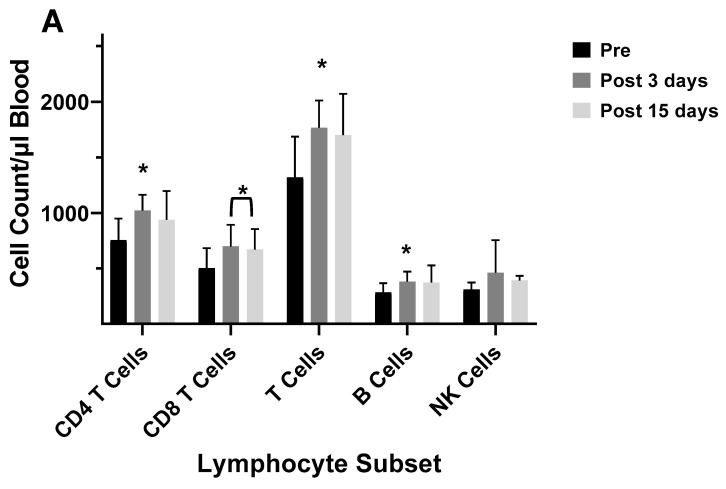
Inclusion of the pandemic strain in the 2010/2011 vaccine resulted in an increase in both CD4^+^ and CD8^+^ T cell numbers in the peripheral blood following the vaccination. This increase persisted until the pandemic strain was removed for the 2017/2018 flu season. Immunophenotyping was performed by the 6-color MultiTest method and samples were run on the FACS Canto flow cytometer with FACS Canto clinical software v2.0 according to BD biosciences protocols. (**A**) Subjects from 2010 vaccination (N = 3). (**B**) Subjects from the 2016 vaccination (N = 2). (**C**) Subjects from the 2017 vaccination (N = 5). Significance was determined by a paired Student’s *t*-Test, “*” significant at *p* < 0.05.

**Figure 9 vaccines-12-00389-f009:**
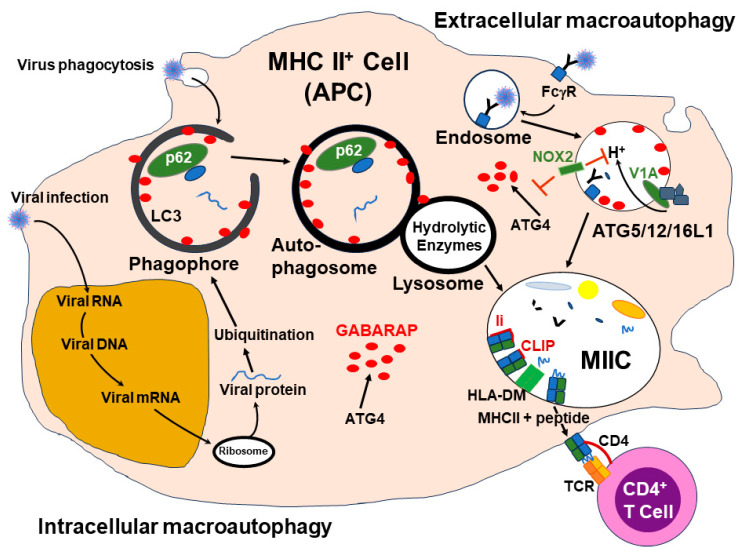
An illustration of flu antigens from either infection or vaccination being processed through the macroautophagy cell machinery for presentation to CD4^+^ T cells. During infection the virus enters the cell, and the RNA genome is incorporated into the DNA in the cell nucleus. The viral genes are then transcribed and translated into proteins. Many of these virus proteins will be ubiquitinated and end up in a phagophore through interaction with LC3 and the autophagy receptor p62 (sequestosome), which is a ubiquitin-binding scaffold protein. This pathway is the same for phagocytized virus vaccine. The autophagy-related gene 8 protein (ATG8) orthologue, GABARAP, is one of several orthologues involved in capture of the viral proteins for breakdown in the lysosome and peptide binding to the MHC II molecule. ATG4 is a family of cysteine proteases that function to process the orthologues from their pro-forms into their active forms. From the lysosome the viral peptides channel into the late-endosomal MHC class II compartment (MIICs) for further processing and loading onto the MHC II molecules. The path via the extracellular TLR receptor does not play a substantial role for flu peptides but intracellular TLR are stimulated by flu nucleotides.

**Figure 10 vaccines-12-00389-f010:**
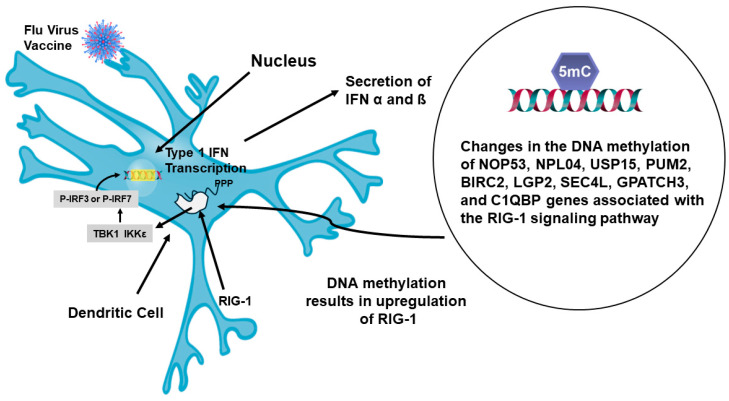
Longitudinal flu vaccination increases methylation of genes associated with the pattern recognition receptor, RIG-1. DNA methylation decreases expression of these elements, and the outcome is increased RIG-1 function. Activation of RIG-1 through a vaccination will lead to triggering of a pathway that will result in phosphorylation of transcription factors IRF3 or IRF7 through TANK-binding kinase 1 (TBK1), an analogue of IKKε. The phosphorylated transcription factors will enter the cell nucleus and promote the transcription of genes for type 1 IFNs. Translation of IFN mRNAs into protein will lead to secretion of IFN-α and IFN-ß from the antigen presenting cell.

**Figure 11 vaccines-12-00389-f011:**
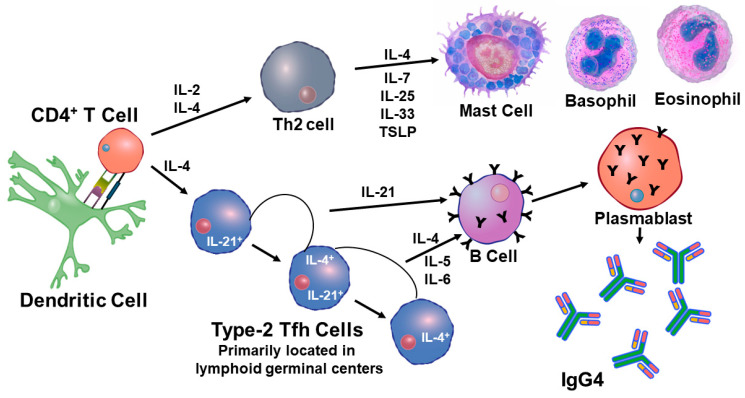
Booster vaccinations skew the immune response toward type 2. APC activation of CD4^+^ T cells with the same epitopes appears to primarily lead to Th2 and Tfh2 phenotypes, which will promote B cell switching from IgG1 to IgG4. This is likely due to higher levels of type 2 cytokine IL-4. The IL-4 and IL-21 will stimulate memory B cells to generate plasmablasts that will secrete IgG4 antibodies. The type 2 responses also include mast cells, basophils, and eosinophils, which are more involved in allergic responses. IgG4 Abs are useful for naturalization but weakly stimulate inflammation due to low affinity to FcγRs.

**Table 1 vaccines-12-00389-t001:** Influenza Vaccine Compositions between the Years 2003 and 2017.

**2003–2004 vaccine**
A/New Caledonia (H1N1), A/Panama (H3N2), and B/Hong Kong
**2005–2006 vaccine**
A/New Caledonia (H1N1), A/California (H3N2), and B/Shanghai
**2006–2007 vaccine**
A/New Caledonia/20/99 (H1N1), A/Wisconsin/67/2005 (H3N2), and B/Malaysia/2506/2004
**2007–2008 vaccine**
A/Solomon Islands/3/2006 (H1N1), A/Wisconsin/67/2005 (H3N2), and B/Malaysia/2506/2004
**2008–2009 vaccine**
A/Brisbane/59/2007 (H1N1), A/Brisbane/10/2007 (H3N2), and B/Florida/4/2006
**2009–2010 vaccine**
A/Brisbane/59/2007, IVR-148 (H1N1), A/Uruguay/716/2007, NYMC X-175C (H3N2), B/Brisbane/60/2008 (3,11)
**2010–2011 vaccine**
A/California/07/2009 (HINI), A/Perth /16/2009 (H3N2), B/Brisbane/60/2008
**2011–2012 vaccine**
A/California/07/2009 (H1N1), A/Perth/16/2009 (H3N2), B/Brisbane/60/2008
**2014–2015 vaccine**
A/California/07/2009 (H1N1), A/Texas/50/2012 (H3N2), B/Massachusetts/02/2012
**2016–2017 vaccine**
A/ California/07/2009 X-179A (H1N1), A/Hong Kong/4801/2014 X-263-B (H3N2), B/Brisbane/60/2008 (B Victoria Lineage)
**2017–2018 vaccine**
A/Michigan/45/2015 (H1N1) pdm09-like virus, A/Hong Kong/4801/2014 (H3N2)-like virus, B/Brisbane/60/2008-like virus (B/Victoria lineage), B/Phuket/3073/2013-like virus (B/Yamagata lineage)

## Data Availability

Examples of raw flow cytometry data are included in the Appendix A. Further inquiries can be directed to the corresponding author.

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
