# Peer review of "Cellular and Molecular Immunity to Influenza Viruses and Vaccines"

_vaccines, 2024, doi:10.3390/vaccines12040389_

Round 1

Reviewer 1 Report (Previous Reviewer 1)

Comments and Suggestions for Authors

The authors have considered reviewers' comments and recommendations. 

Author Response

Thank you for the second review and for your acceptance of our changes.

Reviewer 2 Report (Previous Reviewer 2)

Comments and Suggestions for Authors

The manuscript was extensively  revised, including figure 1 and figure 2 made the review much more easily to read. I agree that including your previous data is helpful in this review. My questions and comments have been addressed adequately. 

Author Response

Thank you for doing a second review and the acceptance of the changes that we made. We revised figure 1 to more closely match the text of the introduction and figure 2 had a minor modification.

Best Regards,

Jane

Reviewer 3 Report (Previous Reviewer 3)

Comments and Suggestions for Authors

Review of "Cellular and Molecular Immunity to Influenza Viruses and Vaccines"

This manuscript is a timely review of current thinking concerning the problems associated with influenza virus vaccinations.  The long-observed issue of "original antigenic sin" or immune-imprinting that interferes with developing highly effective immune defenses against new variants or pandemic strains of influenza viruses has been described many times.  In addition, this problem likely becomes worse as patients age and are repeatedly exposed to newer strains of virus.  There may very well be a genetic component(s) that further contributes to poor immune responses to evolving influenza strains as the author's data suggest.   

I especially liked the use of the "cartoon-style" diagrams to help illustrate some of the concepts presented here.  I especially appreciated the anti-idiotype immune network cascade diagram as I have always struggled with how this might work to regulate antibody responses.  I would like to point out that Figure 9 has a small mistake.  The label "viral DNA" entering the nucleus should read viral RNA since genomic influenza viral RNA enters the nucleus for mRNA transcription. 

Overall, this is a good and informative review of influenza vaccination issues and provides a nice discussion of how immune imprinting may develop and how it could provide clues to getting around this problem.  The authors point out that a similar issue may be developing with the COVID-19 vaccines upon repeated immunizations for variants that arise.    

Author Response

Thank you for doing a second review of our manuscript and the acceptance of our changes. We have changed figure 9 in accordance with your suggestion. The figure legend did not need to be changed.

Best Regards,

Jane

This manuscript is a resubmission of an earlier submission. The following is a list of the peer review reports and author responses from that submission.

Round 1

Reviewer 1 Report

Comments and Suggestions for Authors

The paper comprehensively reviews the immune response to influenza infections and vaccinations. It discusses the concept of "original antigenic sin," the theories of antibody generation, and the comparison between immune responses to live infections and inactivated flu vaccinations. The study delves into the roles of B cells, T cells, NK cells, and macrophages and the mechanisms of antigen presentation and epitope focus for vaccine development.

Drawbacks:

1. While the paper extensively covers the cellular and molecular mechanisms, there could be a more detailed discussion on the clinical implications of these findings, particularly in the context of vaccine design and effectiveness.

2. The paper could benefit from including a wider range of studies, especially recent ones, to support its conclusions and provide a broader perspective on the rapidly evolving field of immunology and vaccine technology.

3. Although the paper touches on developing a universal flu vaccine, it could further explore emerging technologies, such as mRNA vaccines, which have gained prominence due to the COVID-19 pandemic.

4. The paper lacks a review methodology.

5. Some figures are unavailable.

Recommendations:

1. It is recommended to include a section dedicated to the clinical implications of the discussed immune mechanisms, highlighting how these findings can influence vaccine development and strategies for flu prevention.

2. The authors should consider incorporating findings from more recent studies to ensure the review captures the latest advancements and perspectives in the field.

3. A detailed discussion on the potential and challenges of novel vaccine technologies, including mRNA vaccines, could enrich the paper and make it more relevant to current and future influenza vaccine development efforts.

4. The review methodology section should be included.

Author Response

please see the attached responses

Reviewer 2 Report

Comments and Suggestions for Authors

This review on immune responses on human seasonal influenza infection and vaccination, study and compare human  antibody and T cells  responses against influenza infection or/and vaccination. The topic is of great interests because of the yearly outbreaks of seasonal influenza epidemic and implementation of worldwide flu vaccination programme. My comments are:  this manuscript be added more in-depth analysis and discussion on previous findings or theories

For instance, to the theory of “original antigenic sin” and the “anti-idiotype antibodies in the human immune responses to influenza vaccination or infection, make more discussion on the biological significance of these theories to seasonal influenza infection or yearly vaccination; Secondly, the contents for antibody production, T/B cell (memory cells) responses to infection of vaccination may be more valuable to the reader by including novel immunology findings. The data presented in the manuscript are not quite representative, some of them are old data from more twenty year ago (year 2003-2005).

Author Response

please see the attached responses

Reviewer 3 Report

Comments and Suggestions for Authors

Review of "Cellular and Molecular Immunity to Influenza Viruses and Vaccines"

This manuscript is a combination of a review and additional experimental data from the authors utilizing surface plasmon resonance (SPR) to measure human IgG binding to various influenza vaccines (2006/2007, 2007/2008 and 2008/2009) as well as flow cytometric assays to determine T, B or NK cell subsets in humans pre- and post-flu vaccinations. 

The experimental data in the manuscript in general support the well-described property of "original antigenic sin/immune imprinting/immune senority".  The number of patients are quite low in some of the groups in Figures 4 & 5.  It should be pointed out that not all studies involving influenza vaccination support the "original antigenic sin" observation which is reviewed in Yewdell and Santos "Original Antigenic Sin: How Original? How Sinful?  Cold Spring Harbor Perspect. Med. 2021 11(5)

The review portion of the manuscript can be difficult to follow in some sections and could benefit from some illustrations in several places.  For example, Figure 1 in your reference #16 Nakamura et al has a nice illustration of idiotypes and anti-idiotypes that make your text more readily understandable to the average reader.

Figure 1 in your reference #20 nicely illustrates the overall concept of OAS/imprinting/senority and would complement your text very well.

Figure 6 in PNAS vol 109(23) 9047-9052 2012 has a nice illustration of the proliferation of memory B cells in seasonal versus pandemic influenza. 

I am not suggesting that you use these illustrations without permission (of course) but simply as an example to assist the reader in visualizing the points or concepts that you are trying to get across. I suspect you could improve on the suggested illustrations or have better ideas. I found that I had difficulty following some of your brief explanations in the text and had to go to your references to clarify your points. 

Minor point: Line 354- 5 days instead of one week? See legend in Figure 3C 

Author Response

please see the attached responses
